# Ponciri Fructus Immatarus Sensitizes the Apoptotic Effect of Hyperthermia Treatment in AGS Gastric Cancer Cells through ROS-Dependent HSP Suppression

**DOI:** 10.3390/biomedicines11020405

**Published:** 2023-01-30

**Authors:** Chae Ryeong Ahn, Hyo In Kim, Jai-Eun Kim, In Jin Ha, Kwang Seok Ahn, Jinbong Park, Young Woo Kim, Seung Ho Baek

**Affiliations:** 1Department of Science in Korean Medicine, Graduate School, Kyung Hee University, Seoul 02447, Republic of Korea; 2Department of Surgery, Beth Israel Deaconess Medical Center, Harvard Medical School, Boston, MA 02215, USA; 3College of Korean Medicine, Dongguk University, 32 Dongguk-ro, Ilsandong-gu, Goyang-si 10326, Republic of Korea; 4Korean Medicine Clinical Trial Center (K-CTC), Korean Medicine Hospital, Kyung Hee University, Seoul 02447, Republic of Korea; 5College of Korean Medicine, Kyung Hee University, 24 Kyungheedae-ro, Dongdaemun-gu, Seoul 02447, Republic of Korea

**Keywords:** hyperthermia, gastric cancer, Ponciri Fructus Immaturus, heat shock proteins, reactive oxygen species, combination therapy

## Abstract

Gastric cancer has been associated with a high incidence and mortality, accompanied by a poor prognosis. Given the limited therapeutic options to treat gastric cancer, alternative treatments need to be urgently developed. Hyperthermia therapy is a potentially effective and safe treatment option for cancer; however, certain limitations need to be addressed. We applied 43 °C hyperthermia to AGS gastric cancer cells combined with Ponciri Fructus Immaturus (PF) to establish their synergistic effects. Co-treatment with PF and hyperthermia synergistically suppressed AGS cell proliferation by inducing extrinsic and intrinsic apoptotic pathways. Additionally, PF and hyperthermia suppressed factors related to metastasis. Cell cycle arrest was determined by flow cytometry, revealing that co-treatment induced arrest at the G2/M phase. As reactive oxygen species (ROS) are critical in hyperthermia therapy, we next examined changes in ROS generation. Co-treatment with PF and hyperthermia increased ROS levels, and apoptotic induction mediated by this combination was partially dependent on ROS generation. Furthermore, heat shock factor 1 and heat shock proteins (HSPs) were notably suppressed following co-treatment with PF and hyperthermia. The HSP-regulating effect was also dependent on ROS generation. Overall, these findings suggest that co-treatment with PF and hyperthermia could afford a promising anticancer therapy for gastric cancer.

## 1. Introduction

Gastric cancer has a poor prognosis owing to limited treatment options [1]. According to GLOBOCAN 2020, the incidence rate of stomach cancer ranks fourth, while the mortality rate ranks third among all cancers [2]. Most gastric cancers are classified as adenocarcinomas, and chemotherapy is essential because more than 50% show distant metastatic lesions at onset [3]. The 5-year survival rate of patients with gastric cancer is approximately 30%, and the survival rate decreases with tumor metastasis [4]. The standard first-line chemotherapy for metastatic gastric cancer is 5-fluorouracil (5-FU) alone, and platinum analog combination therapy is generally recommended as first-line systemic therapy [5]. Moreover, recent guidelines suggest triplet therapy with fluoropyrimidine plus a doublet of platinum-based drugs. Although improvement in the survival rate remains minimal when compared with that of double therapy, safety profiles were found to be improved in terms of toxicity [6]. However, these therapies are well-known to induce side effects. Therefore, the development of alternative therapies is warranted.

Natural products have several advantages, such as multiple target mechanisms and limited side effects [7]. The immature fruit of Poncirus, or Ponciri Fructus Immaturus (PF), is the unripe fruit of the tangerine tree (*Poncirus trifoliata* Rafinesque), which is used as a medicinal herb in Asian countries. Studies have reported its anticancer effects in promyelocytic leukemia [8], colon cancer [9], hepatocellular carcinoma [10,11], oral cancer [12], breast cancer [13], and melanoma [14]. In addition, PF has a good safety profile; it is considered non-toxic when compared with other natural products. This is a considerable advantage, given that although several natural products are known to exert anticancer properties, some have been deemed toxic for long-term use [15].

In the present study, we combined PF and hyperthermia to promote cancer cell death. Hyperthermia can induce several physiological responses via high-temperature stimulation. In particular, hyperthermia can induce cancer cell death [16]. Combined with conventional chemotherapy, hyperthermia treatment can potentiate these effects [17,18,19,20]. Furthermore, hyperthermic chemotherapy can suppress side effects in patients with gastric cancer undergoing long-term treatment [21], and combined treatment during cytoreductive surgery suppresses peritoneal diseases [22]. In addition, hyperthermia has been used to prevent recurrence in patients with gastric cancer [23].

Previously, we have demonstrated that combining hyperthermia and certain natural products could synergistically induce apoptosis in cancer cells. Cinnamaldehyde, a component of cinnamon, synergistically induced the apoptosis of A549 non-small lung cancer cells or ACHN renal cell carcinoma cells with 43 °C hyperthermia through regulation of reactive oxygen species (ROS) and mitogen-activated protein kinases (MAPKs) [24,25]. Herein, we evaluated the synergistic cancer cell death effect induced by PF and hyperthermia in AGS cell lines and examined potential underlying mechanisms.

## 2. Materials and Methods

### 2.1. Reagents

PF (Kwangmyeongdang Medicinal Herbs Co. Ltd. Ulsan, Republic of Korea) was thoroughly ground and homogenized using a homogenizer. The extract was then obtained by soaking for 24 h at room temperature in 70% EtOH. The resulting extract was lyophilized, concentrated under low pressure, and filtered (pore size: 5 μm). Dimethyl sulfoxide (DMSO) (Samcheon Chemical, Seoul, Korea) solutions of 100 and 200 μg/mL were prepared. All solutions were stored at 4 °C until use.

### 2.2. Liquid Chromatography (LC)-Mass Spectrometry (MS) Analysis

Chromatographic analysis using ultra-performance liquid chromatography-electrospray ionization/quadrupole-time-of-flight high-definition mass spectrometry/mass spectrometry (UPLC-ESI-QTOF-MS/MS) was performed to identify the chemical components in the ethanol extract. The extract was shaken in 50% methanol using a vortex mixer for 30 s and sonicated for 10 min. The supernatants were filtered through a 0.2 μm hydrophilic polytetrafluoroethylene syringe filter (Thermo Scientific). The filtrate was diluted to 100 mg/mL and transferred to an LC sample vial before use. The liquid chromatography–mass spectrometry system consisted of a Thermo Scientific Vanquish UHPLC system (Thermo Fisher Scientific, Sunnyvale, CA, USA) with a Poreshell EC-C18 column (2.1 mm × 100 mm, 2.7 μm; Agilent, Santa Clara, CA, USA) and a Triple TOF5600+ mass spectrometer system (QTOF MS/MS, SCIEX, Foster City, CA, USA).

The QTOF MS was equipped with an electrospray ionization (ESI) source in positive and negative ion modes and was used to complete the high-resolution experiment. The elution program for UHPLC separation employed 0.1% formic acid in water as eluent A and methanol as eluent B, as follows: 0–10 min, 5% B; 10–30 min, 5–80% B; 30–31 min, 80–100% B; 31–35 min, 100% B; and equilibration with 5% B for 4 min at a flow rate of 0.3 mL/min. The column temperature was 25 °C, and the autosampler was maintained at 4 °C. The injection volume for each sample solution was 5 μL. Data acquisition and processing for qualitative analysis were conducted using Analyst TF 1.7, PeakVeiw2.2, and MasterView (SCIEX, Foster City, CA, USA). The MS/MS data for qualitative analysis were processed using PeakView and MasterView software to screen for probable metabolites based on accurate mass and isotope distributions.

### 2.3. Cell Culture

The AGS gastric cancer cell line was obtained from the Korean Cell Line Bank (Seoul, Korea). The cells were maintained in an incubator at 37 °C with humidified air containing 5% CO_2_ and RPMI1640 media supplemented with 10% heat-inactivated fetal bovine serum (Gibco, Grand Island, NY, USA) and 1% Pen-Strep (10,000 U/mL) (Gibco, Grand Island, NY, USA).

### 2.4. Hyperthermia Treatment

Briefly, AGS cells were seeded in 6-well plates (0.3 × 10⁶ cells) and suspended in 3 mL of media before incubation in a water bath, with temperature control at 37 or 43 °C for 30 min (unless indicated otherwise). PF was added to samples 1 h earlier, at the indicated concentrations.

### 2.5. MTT Assay

To measure cell growth following exposure to high-heat PF, the MTT assay was performed. AGS cells (1 × 10^4^ cells/mL) were seeded in 96-well plates and allowed to attach overnight. An untreated group served as the control, and each group had three wells. Following cell fixation, different concentrations of PF (100 and 200 μg/mL) were added to the plate. The plate was then incubated for 1 h at 37 °C in a humid environment with 5% CO_2_ after adding various PF concentrations. Subsequently, the plates were incubated in a temperature-controlled water bath set at 37 or 43 °C for 30 min. After 48 h, 20 μL of MTT (2 mg/mL in phosphate-buffered saline (PBS)) (AMRESCO, Solon, OH, USA) was added, and each well was incubated for an additional 2 h. The culture media were then removed and and cells were lysed in 100 μL DMSO. The absorbance was measured at 570 nm using an automated spectrophotometric plate reader. The percentage of relative cell viability was standardized against that of untreated controls. Compusyn (ver. 1.0) was used to calculate the synergistic effects mediated by co-treatment with PF and hyperthermia.

### 2.6. Trypan Blue Assay

After Trypan blue (Sigma-Aldrich, St. Louis, MO, USA) staining (0.4%, 1:1 dilution in the cell-containing PBS), cell viability was measured using a hemocytometer. Briefly, AGS cells (0.3 × 10⁶) were seeded in 6-well plates, followed by treatment with PF for 1 h and hyperthermia (30 min). The cells were collected, diluted with PBS (1:4) after a post-treatment incubation period of 24 h, stained, and counted. The cell survival rate was calculated as follows:Cell survival rate = Viable cell count/Total cell count × 100%

### 2.7. Morphology Assay

Cell proliferation was examined using a morphological assay. AGS cells were plated at a density of 0.3 × 10⁶ cells per well in 6-well plates. Following attachment, cells were treated with 200 μg/mL PF for 1 h, followed by 30 min incubation at 37 or 43 °C. After 24 h, cells were observed and photographed under a microscope (CX-40; Olympus, Tokyo, Japan).

### 2.8. Wound Healing Assay

AGS cells were triplicated in 6-well plates at a density of 0.3 × 10⁶ cells per well and maintained at 37 °C. On reaching the desired confluency, a thin scratch was placed on each well using a yellow pipette tip. A microscope (CX-40, Olympus, Tokyo, Japan) was used to capture images (0 h). The cells were rinsed with PBS, the culture medium was removed after 24 h, and images were obtained (24 h). Narrowed gap distances were measured after 24 h incubation and normalized against the baseline control at 0 h.

### 2.9. Colony Formation Assay

Cells were seeded on a 6-well plate at a density of 400 cells/well and incubated overnight. The cells were incubated at 37 or 43 °C for 30 min before exposure to 200 μg/mL PF for 1 h. After one week, the cells were stained for 10 min at room temperature using a crystal violet solution (Sigma-Aldrich, St. Louis, MO, USA), followed by rinsing with PBS. Subsequently, colonies were observed under a microscope (CX-40; Olympus, Tokyo, Japan).

### 2.10. Western Blot Analysis

Briefly, AGS cells were extracted and protein concentrations were calculated. Equal amounts of the lysates separated by sodium dodecyl-polyacrylamide gel electrophoresis (SDS-PAGE) were transferred to a polyvinylidene difluoride (PVDF) membrane, and the membrane was then blocked at room temperature with 1× TBS containing 0.1% Tween 20 and 5% skim milk. After blocking, the matched primary antibodies (1:3000) were applied to the membranes, including the following primary antibodies at 4 °C overnight: anti-caspase-3, anti-survivin, anti-heat shock protein (HSP) 27, anti-HSP70, anti-HSP90, anti-caspase-8, anti-caspase-9, anti-p-extracellular signal-regulated kinase (ERK) (Thr202/Tyr204), anti-ERK, anti-p-Jun N-terminal kinase (JNK) (Thr183/Tyr185), anti-JNK, anti-p-p38 (Thr180/Tyr182), anti-p38 (Cell Signaling Technology), anti-β-actin, anti-Bcl-2, anti-Bcl-xL, anti-cyclin D1, anti-vascular endothelial growth factor (VEGF), anti-matrix metallopeptidase (MMP) 9, anti-MMP2, anti-Cyclin B1 (Santa Cruz Biotechnology, Inc.), anti-cleaved caspase (Genetex), anti-heat shock factor 1 (HSF1), and anti-pHSF1 (Abcam, Inc.). After washing three times, membranes were incubated for 1 h at room temperature with diluted anti-rabbit or anti-mouse IgG secondary antibodies (Santa Cruz Biotechnology, Inc., 1:1000). Between each step, the blots were thrice rinsed with 1 TBS-T buffer for 10 min. The membranes were detected using an enhanced chemiluminescence (ECL) kit (Millipore, Billerica, MA, USA).

### 2.11. Annexin V Apoptosis Assay

An apoptosis assay was performed using an Annexin V-FITC Detection Kit (Cat. No.: LS-02-100). PF and heat treatment were applied to AGS cells (0.3 × 10⁶ cells/well) in a 6-well plate for 24 h. Then, the cells were collected and stained with Annexin V-FITC in 1× cold binding buffer for 15 min at room temperature, protected from light. After removing the supernatant by centrifugation, propidium iodide (PI) staining was performed using 1× *g* cold binding buffer. Flow cytometry was used to analyze Annexin V staining to detect apoptotic cells.

### 2.12. Cell Cycle Analysis

Co-treatments were applied to AGS cells (0.3 × 10⁶ cells/well) in 6-well plates for 24 h. Cells were then collected, frozen in 70% ice-cold EtOH overnight, washed in PBS, and resuspended in PBS containing 1 mg/mL PI and 10 mg/mL RNase A in a dark environment for 10 min to determine the cell cycle phase. Flow cytometry was used to determine the cell cycle.

### 2.13. ROS Analysis

The ROS assay was performed using the 2′,7′-dichlorofluorescin diacetate (H2DCFDA) reagent (Invitrogen ™ D399). AGS cells (0.3 × 10⁶ cells/well) were attached to a 6-well plate, and co-treatments were applied for 4 h. After collecting cells, 10 μM reagent was added, followed by incubation for 40 min at 37 °C protected from light. The ROS signals were observed using flow cytometry.

### 2.14. Statistical Analysis

Numerical values are presented as the mean ± standard deviation (SD). Statistical significance was determined using the Student’s unpaired *t*-test, with *p* values < 0.05 deemed statistically significant.

## 3. Results

### 3.1. UPLC-ESI-QTOF-MS/MS Analysis for Identification of Chemical Components in PF

The ethanol extract was subjected to UPLC-ESI-QTOF-MS/MS to determine the chemical profile and identify extract constituents. The extracted ion chromatogram revealed 30 known components, including naringin (J14) and poncirin (J21), the two main compounds of *Poncirus trifoliata* Rafinesque (Figure 1). The detected peaks are listed in Table 1.

### 3.2. Co-Treatment with PF and 43 °C Hyperthermia Synergistically Inhibits AGS Cell Proliferation

Using MTT assays, we examined the effects of co-treatment with PF and hyperthermia at 37 and 43 °C. At the same dose of PF (200 µg/mL), co-treatment with 43 °C more significantly reduced AGS cell viability than co-treatment with 37 °C (Figure 2a). The synergistic effect of PF and hyperthermia was determined by calculating the combination index (Figure 2b). Based on morphological observations, co-treatment with the same PF concentration and increasing temperature suppressed cell growth (Figure 2c). Examining crystal violet-stained AGS cells, co-treatment with PF and 43 °C markedly reduced colony formation when compared with co-treatment with PF and 37 °C (Figure 2d). A further cell migration assay showed that co-treatment with PF and hyperthermia inhibited cell migration (Figure 2e). Overall, these findings suggested that co-treatment with PF and hyperthermia could exert an anti-proliferative effect against AGS cells.

### 3.3. Co-Treatment with PF and 43 °C Hyperthermia Synergistically Induces Apoptotic Cell Death in AGS Cells

Next, we confirmed the expression levels of factors related to apoptosis, proliferation, metastasis, and angiogenesis to verify the synergistic mechanism of action induced by co-treatment with PF and hyperthermia. PF treatment at 43 °C dose-dependently increased the activated forms of caspase 3, caspase 8, and caspase 9, markers for programmed cell death [26]. However, this result was not observed under normothermic (37 °C) conditions. In addition, co-treatment with PF and 43 °C dose-dependently decreased the expression levels of anti-apoptotic members of the B-cell lymphoma (Bcl)-2 family, including Bcl-2, Bcl-xL, and survivin [27] (Figure 3a). Furthermore, co-treatment with PF and hyperthermia inhibited the metastatic potential of AGS cells, as indicated by suppressed levels of VEGF, MMP-2, and MMP-9 (Figure 3b). As shown in Figure 3c, co-treatment with PF and 43 °C enhanced Annexin V-related apoptosis in AGS cells when compared with that induced by 43 °C hyperthermia alone or PF with normothermia. The synergistic effect of co-treatment with PF and hyperthermia was dose-dependent, demonstrating that the ratio of apoptotic cells (35.49%) increased by nearly 6-fold at the highest PF dose (6.74%) and 43 °C (9.03%).

### 3.4. Co-Treatment with PF and 43 °C Hyperthermia Induces Cell Cycle Arrest in AGS Cells

Cell cycle arrest is closely related to the induction of cellular apoptosis [28] and is frequently employed as a therapeutic target of anticancer drugs [29,30]. We performed flow cytometry to determine whether co-treatment with PF and hyperthermia impacts cell cycle arrest. Co-treatment with PF and 43 °C hyperthermia arrested the cell cycle in the G2/M phase (Figure 4a). Treatment with PF at 43 °C hyperthermia significantly reduced the expression of cyclin B1, corroborating the induction of cell cycle arrest in AGS cells at the G2/M phase (Figure 4b).

### 3.5. Co-Treatment with PF and 43 °C Hyperthermia Synergistically Increases ROS Generation and Subsequent Apoptosis in AGS Cells

ROS signaling is frequently used as a target to induce cancer cell death [31]. Accordingly, we examined the underlying mechanism through which co-treatment with PF and hyperthermia can induce AGS cell death, identifying the role of ROS in the pro-apoptotic effect of combination therapy. Based on flow cytometric analysis (Figure 5a), co-treatment with PF and 43 °C hyperthermia significantly increased ROS levels (panel 4) when compared with PF treatment at 37 °C (panel 3). Next, we pre-treated cells with N-acetylcysteine (NAC), a free radical scavenger used as a ROS inhibitor [32]. NAC pre-treatment nullified the effect of PF and hyperthermia on ROS generation (Figure 6). We then examined whether or not ROS generation played a role in PF/hyperthermia-induced AGS apoptosis. Based on Annexin V staining results, co-treatment with PF and hyperthermia failed to induce apoptosis in AGS cells in the presence of NAC, suggesting that ROS generation played a key role in chemotherapy (Figure 5b).

### 3.6. Co-Treatment with PF and 43 °C Hyperthermia Synergistically HSP via ROS Generation in AGS Cells

HSPs are highly conserved molecular chaperones that contribute to protein homeostasis, transport, and signal transduction. In addition, HSPs play an essential role in protecting cells from stress and the degradation of severely damaged proteins [33,34]. As shown in Figure 6a, HSP27, 70, and 90 were overexpressed in AGS cells incubated at 43 °C hyperthermia. Treatment with PF markedly decreased HSP expression under both normothermia and hyperthermia. HSF1 is activated by stress factors such as heat shock to protect the proteome. Upon phosphorylation, HSF1 acts as a transcription factor, inducing several downstream pathways, including the synthesis of HSPs [35]. HSF1 is overexpressed in cancer cells and participates in the migration, invasion, and proliferation of malignant tumors [36]. Hyperthermia treatment at 43 °C for 1 h can induce phosphorylation of HSF1 [37]. Herein, we observed that co-treatment with PF prevented HSF1 phosphorylation, even after 6 h of hyperthermia incubation (Figure 6b). HSF1 phosphorylation is closely related to the MAPK family [38,39,40]. As shown in Figure 6c, co-treatment with PF could prevent phosphorylation of MAPKs, including JNK, p38, and ERK, in correlation with suppressed HSF1 phosphorylation. However, NAC pre-treatment could reverse the suppressed expression of HSP27 and 70 induced by PF co-treatment, suggesting that the co-treatment effects partially depended on ROS generation. In the absence of ROS release, co-treatment with PF and hyperthermia may fail to sufficiently interrupt the heat shock-induced self-protection system in AGS cells. In addition, ROS generation was critical for apoptosis induction, and NAC pre-treatment restored caspase-3 expression (Figure 6d).

## 4. Discussion

It is estimated that more than one million new cases of gastric cancer were reported in 2020. Globally, males are almost twice as likely to be diagnosed with gastric cancer than females, with the highest incidence in East Asian countries [41]. Adenocarcinoma is the most frequently observed gastric cancer and remains the most lethal malignancy [3], despite efforts to develop superior treatment options. Therefore, identifying effective therapeutic strategies to combat gastric cancer remains a critical challenge.

PF is a medicinal herb that is widely used in oriental medicine. In traditional Korean medicine, PF was primarily used to relieve indigestion and constipation and lacked toxic effects [42]. Numerous clinical and experimental studies have reported that PF can exert various effects on inflammation [43] and cardiovascular diseases [44]. Additionally, PF was found to induce beneficial effects in several cancer types, including leukemia [8], colon cancer [9], hepatocellular carcinoma [10,11], oral cancer [12], triple-negative breast cancer [13], and melanoma [14]. Although its anticancer effects are well-established in several cancers, the potential role of PF in gastric cancer remains unexplored.

The treatment and prognosis of gastric cancer are dependent on the cancer stage, typically evaluated using the American Joint Committee on Cancer tumor-node-metastasis (TNM) system [4,45]. Surgery is the first-line treatment for all stages of gastric cancer [46]. Chemotherapy is one of the most frequently used strategies to treat gastric cancer. Various attempts have been made to maximize the therapeutic effects of chemotherapy. However, chemotherapy-induced side effects remain a considerable challenge; thus, research is ongoing to minimize side effects while retaining anticancer effects. Therefore, hyperthermia is a promising treatment option in this field. Hyperthermia is frequently combined with chemotherapy, affording a synergistic effect [47,48]. In addition, hyperthermia therapy has been used to treat gastric cancer, exhibiting several advantages when combined with chemotherapy [21,49]. Hyperthermia treatment can potentially kill tumor cells by damaging proteins and structures while causing minimal damage to normal tissues [50]. Consistent with our previous reports [24,25], co-treatment with herbal combinations and hyperthermia can afford synergistic anticancer effects [51,52]. In the present study, we attempted to verify the synergistic effect of co-treatment with PF and hyperthermia in the gastric cancer cell line AGS. PF treatment reduced the cell viability of AGS cells down to around 80%, while hyperthermia alone did not affect the cell viability. However, when combined, PF and hyperthermia co-treatment exerted a significant synergism on AGS cell death. Thus, we concluded that co-treatment with PF and hyperthermia could suppress AGS cell proliferation, as determined by assessing cell viability, morphology, and metastasis (Figure 2).

Among several types of programmed cell death, apoptosis is considered the main pathway. Cleavage of caspase 3 is well known as the last step of programmed cell death; therefore, it is commonly used to identify cell death progression [53]. Herein we observed that co-treatment with PF and 43 °C hyperthermia dramatically induced the expression of caspase 3 cleavage, indicating the activation of the apoptotic pathway. Apoptosis is mediated via two distinct pathways: endogenous or intrinsic and extrinsic, induced by death receptors. The mitochondrial response participates in the intrinsic apoptotic pathway. Stimulation of intrinsic mitochondrial apoptosis induces the conversion of caspase 9 to its activated cleavage form. Meanwhile, extrinsic apoptosis induces the cleavage of caspase 8 as a downstream pathway of tumor necrosis factor receptor (TNFR) activation. Intrinsic and extrinsic apoptosis initiate from different pathways, but their common final step involves the cleavage of caspase 3 [54]. We observed that co-treatment with PF and 43 °C hyperthermia decreased expression levels of caspase 8 and 9 (Figure 3a), while standalone treatments only induced mere differences. In addition, the balance between pro-apoptotic and anti-apoptotic members of the Bcl-2 family shifts toward an excessive proportion of anti-apoptotic members, such as Bcl-xL and Bcl-2, thereby initiating intrinsic apoptosis [55]. Our results revealed that co-treatment with PF and hyperthermia regulated the Bcl-2 family (Figure 3a). Accordingly, co-treatment with PF and 43 °C hyperthermia could induce both apoptotic pathways in AGS cells, which was not significantly induced by PF only or hyperthermia alone.

In addition, metastasis is an important issue in cancer treatment, particularly in patients with gastric cancer. In patients with metastatic gastric cancer, chemotherapy is the first-line treatment, and targeted therapy, immunotherapy, or radiation is considered [56]. However, metastasis of gastric cancer mostly results in a poor prognosis. Patients with metastasized gastric cancer exhibit a poor 5-year survival rate, reduced to 6%. The 5-year survival rate of patients with localized gastric cancer is 70%. However, more than one-third of patients are diagnosed at late stages, accompanied by metastasis [57]. Several pathways are known to be involved in cancer metastasis. Herein, we demonstrated that PF could suppress the metastatic markers MMP-2 and MMP-9 [58] at the highest dose of 200 μg. Additionally, co-treatment with PF and hyperthermia at 43 °C further enhanced the suppression of metastatic markers, exhibiting suppression at low doses while also inhibiting other factors such as VEGF and cyclin D1 [59]. Overall, hyperthermia treatment failed to induce notable suppression of these markers, except MMP-9 (Figure 3b). As shown in Figure 3, co-treatment with PF and hyperthermia could synergistically induce AGS cell apoptosis and inhibit metastasis.

In eukaryotic cells, the cell cycle consists of the G1, S, G2, and M phases. Normal cell growth and death are regulated by checkpoints. However, checkpoints fail to delay cell cycle progression when cells are damaged or mutated, resulting in an abnormal cell cycle that allows continuous cell division, the predominant characteristic of cancer [30]. Cyclin B1 and D1 regulate cell mitosis, adhesion, and cell cycle migration, thereby participating in cancer progression and metastasis [60]. Therefore, the cell cycle of cancer cells is a well-established target for anticancer therapies [30]. Our results showed that either PF or hyperthermia induced cell cycle arrest; however, co-treatment with PF and hyperthermia could greatly enhance G2/M arrest (Figure 4a) and suppress cyclin B1 expression (Figure 4b). Based on these findings, we concluded that co-treatment with PF and hyperthermia could induce cell cycle arrest and thus led to AGS cell apoptosis.

The expression of HSPs in cancer is known to induce carcinogenesis, proliferation, migration, and metastasis [61]. Accumulated clinical evidence indicates that HSP27 expression is closely associated with the incidence of gastric cancer [62,63]. HSP70 is also associated with tumor differentiation and metastasis of gastric cancer [64], whereas HSP90 plays a substantial role in the invasion, metastasis, and progression of gastric cancer [65]. In addition, these proteins help cancer cells develop resistance against platinum-based anticancer drugs [66]. Therefore, HSPs are often considered therapeutic targets for gastric cancer treatment [67]. Herein, our results revealed that PF treatment inhibited the hyperthermia-induced increase in HSPs. This finding clarifies the function of PF combined with a hyperthermic environment as a more effective strategy for inducing cell death in AGS cells than treatment at 37 °C (Figure 6a). Interestingly though, and also unexpectedly, PF treatment did not suppress the expression of these heat response markers, except for HSP27 when used alone (Figure 6a). However, PF co-treatment could regulate HSF-1, an upstream target of HSPs, and MAPKs, the downstream pathways of HSP activation (Figure 6b,c).

ROS is considered a crucial pathway in the hyperthermia treatment process. Hyperthermic intraperitoneal chemotherapy, which involves infusion and circulation of chemotherapy after heating the anticancer drug, especially for abdominal cancers, has documented the evident involvement of ROS [68]. HSP27 and HSP70, which are overexpressed under oxidative stress, were found to be associated with tumor metastasis, poor prognosis, and resistance to chemotherapy [69,70]. Herein, we demonstrated that co-treatment with PF and 43 °C hyperthermia could induce ROS generation (Figure 5a), and the induction of apoptosis by this combination therapy was dependent on ROS (Figure 5b). ROS generation was an effect made possible by PF, because PF treatment increased the ROS expression up to 27.87% (vs. 16.28% in untreated cells). On the other hand, hyperthermia failed to increase ROS (Figure 5a). Importantly, ROS generation was a crucial factor for the HSP-regulating effect mediated by PF and hyperthermia co-treatment. In the presence of pre-treatment with ROS scavenger NAC [32], co-treatment with PF and hyperthermia failed to inhibit HSP expression. Moreover, an apoptotic effect was also observed (Figure 6d). These results are consistent with previous reports exploiting ROS as a therapeutic target of natural products [71].

Figure 7 illustrates the findings of the present study. We show that co-treatment with PF and 43 °C hyperthermia inhibits invasion and induces cell death in AGS gastric cancer cells. Notably, the underlying mechanisms involved the ROS-mediated suppression of HSPs. Apoptosis is also dependent on ROS generation and subsequent HSP regulation. To the best of our knowledge, the present study is the first to demonstrate the anticancer effect of hyperthermia when supplemented with PF. Hyperthermia treatment has potential benefits in cancer treatment; however, HSPs respond to heat and activate a protective system. Our results suggest that a precise combination with natural products, such as PF, can overcome these limitations and maximize the effect of hyperthermia treatment.

## 5. Conclusions

The purpose of this study was to examine the efficacy of employing Ponciri Fructus Immaturus (PF) in conjunction with hyperthermia treatment for stomach cancer. Over one million new instances of gastric cancer will be recorded in 2020 alone, and the incidence is highest in East Asian countries. Finding efficient therapy for gastric cancer continues to be difficult, despite attempts to develop new treatments. It has been demonstrated that PF has varied impacts on inflammation and cardiovascular disease, as well as leukemia, colon cancer, hepatocellular carcinoma, oral cancer, triple-negative breast cancer, and melanoma, among others. However, no research has yet been conducted on its impact on gastric cancer. The study demonstrated that the combination of PF and hyperthermia inhibited cell proliferation and promoted apoptotic cell death in the gastric cancer cell line AGS cells. This was observed through cell viability, morphology, and metastasis assays, as well as caspase 3, 8, and 9 cleavage assays. The research indicates that the combination of PF and heat may be an effective treatment for stomach cancer. Further research is needed to confirm the findings in vivo and in human trials before they can be applied to real-world situations.

## Figures and Tables

**Figure 1 biomedicines-11-00405-f001:**
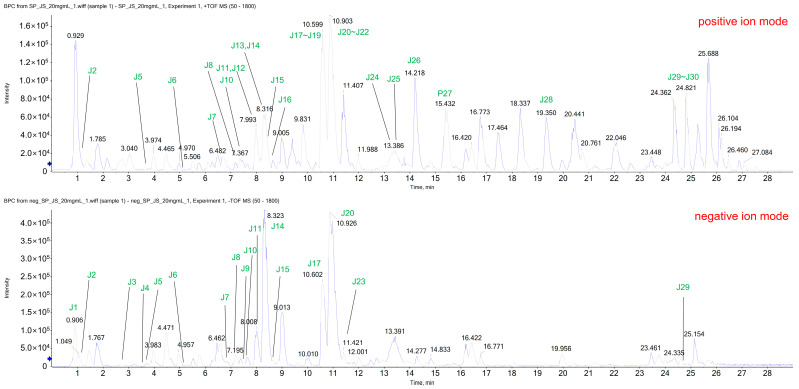
Extracted ion chromatograms (XICs) identified in PF using LC-ESI-QTOF MS/MS analysis in positive and negative ion modes. PF, Ponciri Fructus Immaturus; LC-ESI-QTOF MS/MS, liquid chromatography-electrospray ionization quadrupole time-of-flight mass spectrometry.

**Figure 2 biomedicines-11-00405-f002:**
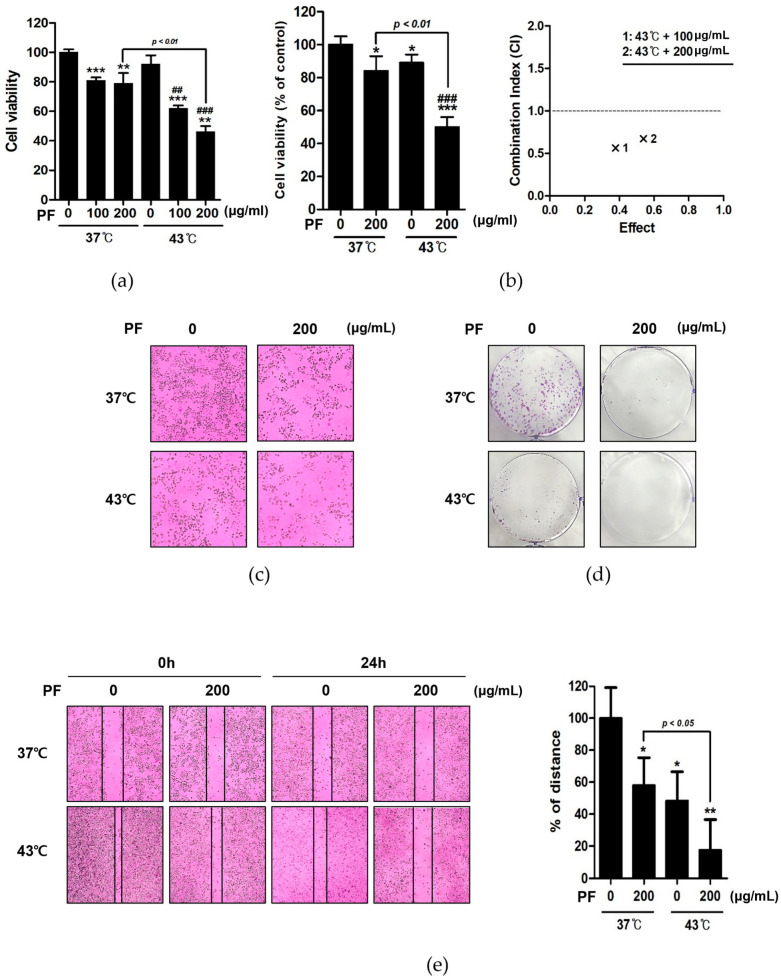
Effect of co-treatment with PF and hyperthermia on AGS cell viability. AGS cells were incubated for 24 h with PF (0, 100, and 200 μg/mL) with or without hyperthermia at 43 °C. (**a**) The MTT assay was used to calculate the percentage of cell viability, and Compusyn software was used to calculate the combination index. (**b**) Trypan blue assay. (**c**) Morphological variations indicating apoptosis were observed using a microscope. (**d**) Crystal violet staining was used for the clonogenic assay. (**e**) Wound healing assay. * *p* < 0.05, ** *p* < 0.01, *** *p* < 0.001 vs. control group; ## *p* < 0.01, ### *p* < 0.001 vs. 43 °C + 0 μg/mL group by Student’s *t*-test. PF, Ponciri Fructus Immaturus.

**Figure 3 biomedicines-11-00405-f003:**
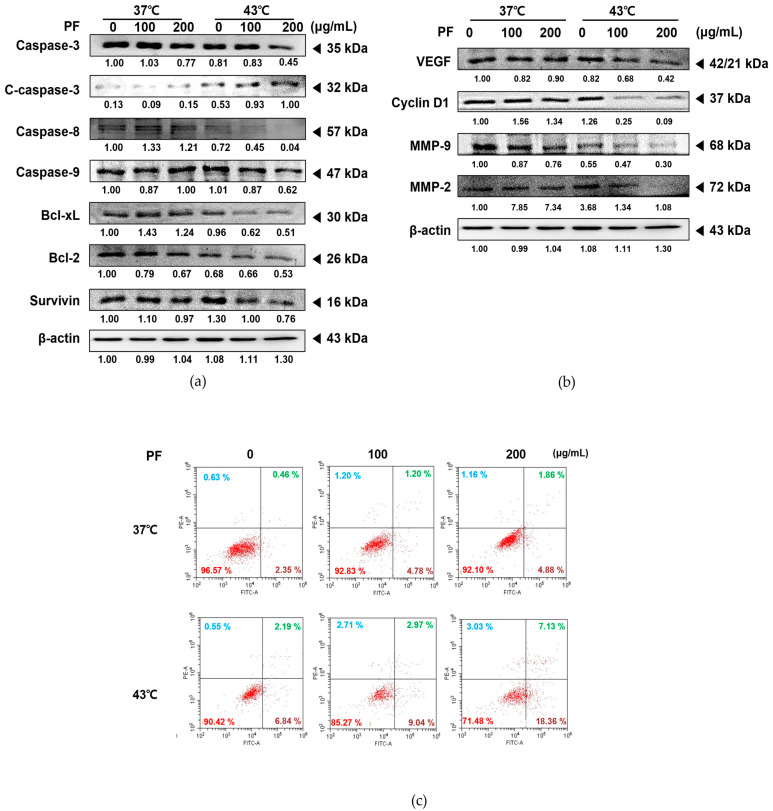
Effect of co-treatment with PF and hyperthermia on apoptosis in AGS cells. PF was applied to AGS cells (0.3 × 10⁶ cells) for 24 h, with or without hyperthermia. Equal volumes of lysates from whole-cell extracts were then subjected to Western blot analysis. Protein expression levels of (**a**) caspase-3, caspase-8, caspase-9 Bcl-2, Bcl-xL, survivin, (**b**) VEGF, MMP-9, and MMP-2 were measured using Western blot assays. β-actin was used as a loading control. (**c**) Annexin V staining was performed to detect apoptotic cells by flow cytometry. MMP-2, matrix metallopeptidase-2; MMP-9, matrix metallopeptidase-9; PF, Ponciri Fructus Immaturus; VEGF, vascular endothelial growth factor.

**Figure 4 biomedicines-11-00405-f004:**
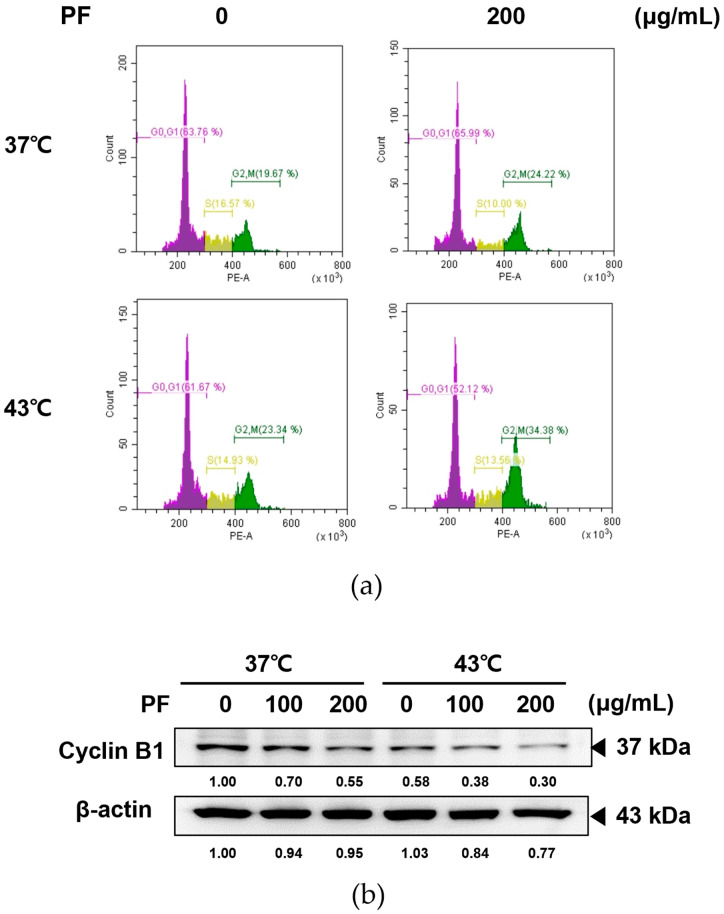
Effect of PF combined with hyperthermia on the cell cycle arrest in AGS cells. After treatment with PF (0, 200 μg/mL) with or without hyperthermia at 43 °C, AGS cells (0.3 × 10⁶ cells) were incubated for 24 h. A flow cytometer was used to analyze the results after apoptosis was identified using Annexin V-FITC and propidium iodide (PI) staining. (**a**) Apoptosis profile and cell cycle profile were analyzed using flow cytometry. (**b**) Western blotting was performed to determine the expression of cyclin B1. β-actin was used as a loading control. PF, Ponciri Fructus Immaturus.

**Figure 5 biomedicines-11-00405-f005:**
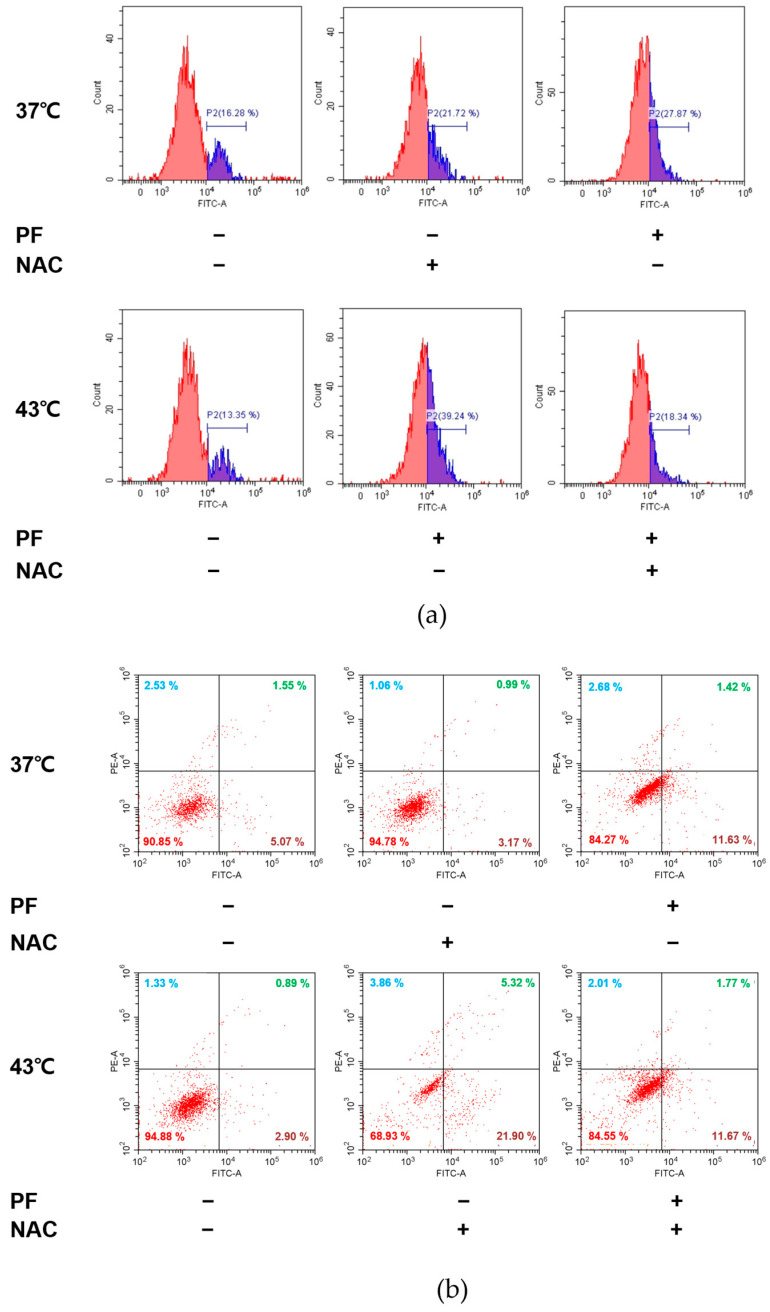
Role of ROS generation in mediating the effect of co-treatment with PF and hyperthermia. Before exposure to PF (0 or 200 µg/mL), with or without hyperthermia of 43 °C, AGS cells were pre-treated with NAC (5 mM) for 1.5 h. (**a**) ROS generation was examined using flow cytometry. (**b**) Annexin V staining was performed. NAC, N-acetylcysteine; PF, Ponciri Fructus Immaturus; ROS, reactive oxygen species.

**Figure 6 biomedicines-11-00405-f006:**
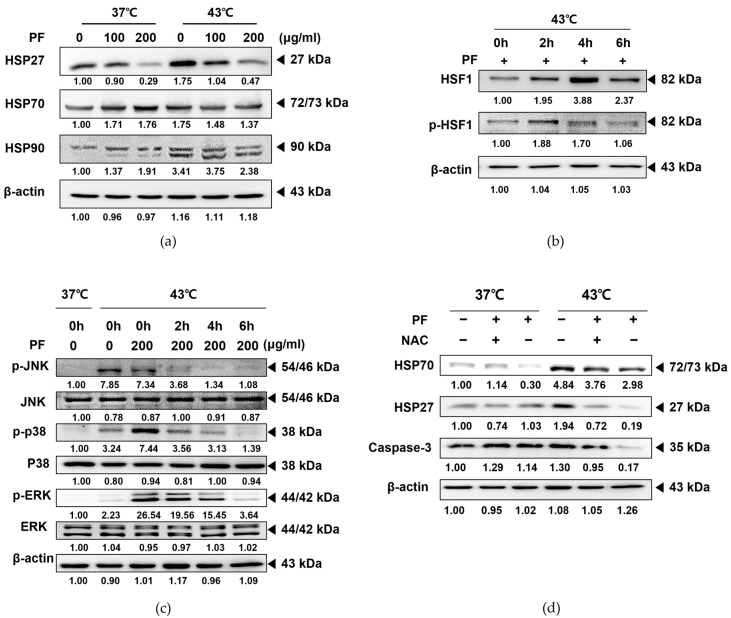
PF potentiates hyperthermia-induced apoptosis in AGS cells by suppressing ROS-mediated heat shock proteins. AGS cells (0.3 × 10⁶ cells) were treated with PF (0 and 200 μg/mL), with or without heat stimulation. Protein expression of (**a**) HSP27, HSP70, and HSP90 was determined by Western blotting. Time course incubation with hyperthermia was performed, and protein levels of (**b**) p-HSF1, HSF, (**c**) p-JNK, JNK, p-p38, p38, p-ERK, and ERK were determined by Western blotting. (**d**) NAC pre-treatment was performed, and protein levels of HSP70, HSP27, and caspase-3 were measured by Western blotting. β-actin was used as a loading control. ERK, extracellular signal-regulated kinase; HSF, heat shock factor; HSP20, heat shock protein 20; HSP70, heat shock protein 70; JNK, Jun N-terminal kinase; NAC, N-acetylcysteine; PF, Ponciri Fructus Immaturus; ROS, reactive oxygen species.

**Figure 7 biomedicines-11-00405-f007:**
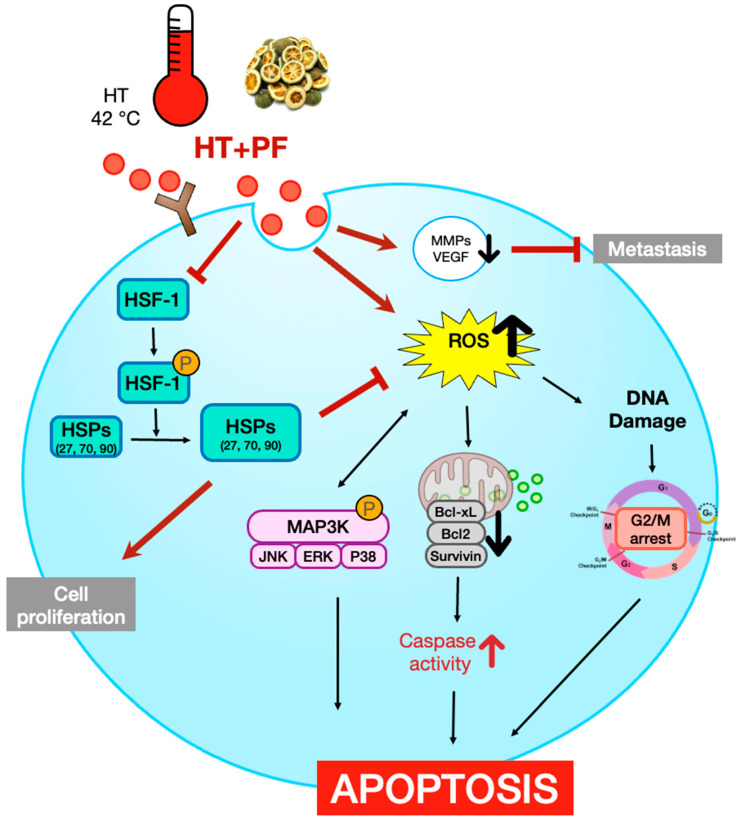
Schematic diagram illustrating the present study. Bcl, B-cell lymphoma; ERK, extracellular signal-regulated kinase; HSF1, heat shock factor 1; HSP, heat shock protein; JNK, Jun N-terminal kinase; MAPK, mitogen-activated protein kinase; MMP, matrix metallopeptidase; NAC, N-acetylcysteine; PF, Ponciri Fructus Immaturus; ROS, reactive oxygen species; TNFR, tumor necrosis factor receptor; VEGF, vascular endothelial growth factor.

**Table 1 biomedicines-11-00405-t001:** Detected peak list from the UPLC-ESI-QTOF-MS/MS analysis of PF.

No.	Name	Formula	Mass (Da)	Expected	Adduct	Found at
				RT (min)		Mass (Da)
J1	Citric acid	C6H8O7	192.027	1.05	[M−H] ^−^	191.01958
J2	Guanosine	C11H15N5O4S	283.09167	1.31	[M+H] ^+^	284.09883
					[M−H] ^−^	282.08411
J3	D-Pantothenic acid	C9H17NO5	219.11067	2.65	[M−H] ^−^	218.10336
J4	1-beta-D-glucopyranosyl-L-tryptophan	C17H22N2O7	366.1427	3.53	[M−H] ^−^	365.13525
J5	Tryptophan	C11H12N2O2	204.08988	3.71	[M+H] ^+^	205.09698
					[M−H] ^−^	203.08254
J6	Quercetin 3-glucosyl-(1->3)-rhamnosyl-(1->6)-galactoside	C33H40O21	772.20621	5.32	[M+H] ^+^	773.2132
					[M−H] ^−^	771.19788
J7	Umbelliferone	C9H6O3	162.03169	6.86	[M+H] ^+^	163.03866
					[M−H] ^−^	161.02466
J8	Rutin	C27H30O16	610.15339	7.20	[M+H] ^+^	611.16014
					[M−H] ^−^	609.14549
J9	Isoquercitrin	C21H20O12	464.09548	7.48	[M−H] ^−^	463.08739
J10	Kaempferol 7-neohesperidoside	C27H30O15	594.15847	7.59	[M+H] ^+^	595.16579
					[M-H] ^−^	593.15067
J11	Narirutin	C27H32O14	580.17921	7.99	[M+H] ^+^	581.18598
					[M−H] ^−^	579.17138
J12	Prunin (Naringenin-7-O-glucoside)	C21H22O10	434.1213	8.00	[M+H] ^+^	435.12884
J13	Naringenin	C15H12O5	272.06847	8.32	[M+H] ^+^	273.0757
J14	Naringin	C27H32O14	580.17921	8.32	[M+H] ^+^	581.18569
					[M−H] ^−^	579.17147
J15	Hesperidin	C28H34O15	610.18977	8.60	[M+H] ^+^	611.19526
					[M−H] ^−^	609.182
J16	Heralenol	C16H16O6	304.09469	9.84	[M+H] ^+^	305.10176
J17	Neoponcirin	C28H34O14	594.19486	10.55	[M+H] ^+^	595.20162
					[M−H] ^−^	595.20162
J18	Heralenol_1	C16H16O6	304.09469	10.58	[M+H] ^+^	305.10191
J19	Heptametoxiflavone	C22H24O9	432.14203	10.59	[M+H] ^+^	433.14889
J20	Isosakuranetin	C16H14O5	286.08412	10.89	[M+H] ^+^	287.09136
J21	Poncirin	C28H34O14	594.19486	10.90	[M+H] ^+^	595.20158
					[M−H] ^−^	593.18695
J22	Heptametoxiflavone_1	C22H24O9	432.14203	10.90	[M+H] ^+^	433.1487
J23	Isosakuranin	C22H24O10	448.13695	11.43	[M−H] ^−^	447.12857
J24	Bergapten	C12H8O4	216.04226	13.33	[M+H] ^+^	217.04951
J25	Isopimpinellin	C13H10O5	246.05282	13.5	[M+H] ^+^	247.06025
J26	Oxyimperatorin (Heraclenin)	C16H14O5	286.08412	14.23	[M+H] ^+^	287.09145
J27	Oxyimperatorin (Heraclenin)_1	C16H14O5	286.08412	15.44	[M+H] ^+^	287.09152
J28	Phellopterin	C17H16O5	300.09977	19.35	[M+H] ^+^	301.10746
J29	Auraptene	C19H22O3	298.15689	24.83	[M+H] ^+^	299.16419
					[M−H] ^−^	297.14938
P25	Limonene or α-Pinene	C10H16	136.1252	24.83	[M+H] ^+^	137.13244

PF, Ponciri Fructus Immaturus; UPLC-ESI-QTOF-MS/MS, ultra-performance liquid chromatography-electrospray ionization/quadrupole-time-of-flight high-definition mass spectrometry/mass spectrometry.

## Data Availability

Not applicable.

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
