# Peer review of "Ponciri Fructus Immatarus Sensitizes the Apoptotic Effect of Hyperthermia Treatment in AGS Gastric Cancer Cells through ROS-Dependent HSP Suppression"

_biomedicines, 2023, doi:10.3390/biomedicines11020405_

Round 1
Reviewer 1 Report
Thank you very much for the nice study done by the authors. I have the following suggestions to improve the manuscript,
Please include the statistical analysis with significance sign for all western blot data.
Please improve the quality and enlarge Figure3c.
3.8. Wound healing assay: please mention how that was measured. Also provide the statistical analysis data along with the figures.
3.10. Western blot analysis: Please provide the antibodies concentrations or dilution used.
Please include a separate conclusion and write based on the results obtained.
Author Response
We sincerely appreciate the effort the reviewer has put on our manuscript. The critical comments have significantly improved our review. Also, the whole text has been thoroughly revised and edited by a native English speaker. We hope our revised manuscript now meets the satisfaction of the reviewer and is acceptable as a publication for Biomedicines.

Reviewer 2 Report
Title
Hyperthermia treatment combined with Ponciri Fructus Immaturus synergistically induces apoptosis in AGS gastric cancer cells via reactive oxygen species-dependent suppression of heat shock proteins
The title is narrated as a conclusion rather than a brief statement about the direction of the study.
Abstract
Gastric cancer has high incidence and mortality rates but shows a poor prognosis.
COMMENT: This is an incorrect statement
Since limited therapy options are available for gastric cancer, there is an unmet need for alternative treatments.
COMMENT: This is an incorrect statement
Hyperthermia therapy is a potentially effective and safe treatment option for cancer, however there are certain limitations that should be overcome. We applied 43°C hyperthermia to AGS gastric cancer cells, together with Ponciri Fructus Immaturus (PF) to study their synergy. PF and hyperthermia co-treatment synergistically suppressed AGS cell proliferation through induction of both extrinsic and intrinsic apoptosis pathways. Also, factors related to metastasis were suppressed by PF and hyperthermia. Cell cycle arrest was determined by flow cytometry, to see that combination treatment induced an arrest at the G2/M phase. Since reactive oxygen species (ROS) is known to be important in hyperthermia therapy, we next studied the changes in ROS generation. ROS was increased by PF and hyperthermia co-treatment, and the apoptotic induction of this combination was partially dependent on ROS. Additionally, heat shock factor 1 and heat shock proteins (HSPs) was notably suppressed by co-treatment of PF and hyperthermia. The HSP-regulating effect was also dependent on ROS generation. Overall, our study suggests combination treatment of PF and hyperthermia as a promising anticancer therapy for gastric cancer.
COMMENT: need to explain what are the positive and negative controls used in this study
In this study, we employed PF and hyperthermia together to promote cancer cell 63 death. Through high-temperature stimulation, hyperthermia causes several physiological 64 responses. In particular, hyperthermia can potentially induce cell death of cancer cells 65 [16]. When combined with conventional chemotherapy, hyperthermia treatment is able to 66 potentiate their effects [17-20]. Even more, reports show that hyperthermic chemotherapy 67 can suppress side effects in gastric cancer patients who receive long-term treatment [21], 68 and combined treatment during cytoreductive surgery suppresses peritoneal diseases 69 [22]. In addition, hyperthermia is also used to prevent recurrence for gastric cancer pa-70 tients [23]. 71
Our previous work also demonstrated the combination of hyperthermia and certain 72 natural products may synergistically induce apoptosis of cancer cells [24,25]. Here, we 73 evaluated the synergistic cancer death effect of PF and hyperthermia in AGS cell lines and 74 investigated the related mechanisms.
COMMENT: the individual effect of hyerthermic therapy and natural product should be explained
The type of natural product (enzyme, extract?) should be included
RESULTS
2.1. UPLC-ESI-QTOF MS/MS Analysis for Identification of Chemical Components in PF
2.2. Combination treatment with PF and 43°C hyperthermia synergistically inhibits cell 89 proliferation of AGS cells
2.3. Combination treatment with PF and 43°C hyperthermia synergistically induces apoptotic 110 cell death in AGS cells
2.4. Combination treatment with PF and 43°C hyperthermia induces cell cycle arrest in AGS 134 cells
2.5. Combination treatment with PF and 43°C hyperthermia synergistically increases reactive 150 oxygen species (ROS) generation and subsequent apoptosis in AGS cells
2.6. Combination treatment with PF and 43°C hyperthermia synergistically inhibits heat shock 169 protein (HSP) through ROS generation in AGS cells
COMMENT:What about the standalone treatment?
3. Materials and Methods
3.1. Reagents
3.2. Liquid chromatography (LC)–mass spectrometry (MS) analysis
3.3. Cell culture
3.4. Hyperthermia treatment
3.5. MTT assay
3.6. Trypan blue assay
3.7. Morphology assay
3.8. Wound healing assay
3.9. Colony formation assay
3.10. Western blot analysis
3.11. Annexin V apoptosis assay
3.12. Cell cycle analysis
3.13. Analysis of ROS
3.14. Statistical analysis
COMMENT: what are the positive and negative controls used in this study-without this, we cannot make a conclusion
Figure 7. Schematic diagram of this study.
COMMENT:
what are the number denotes? The term APOPTOSIS attached to the cell-is this correct?
Author Response

(The authors gave the same response as above.)

Round 2
Reviewer 2 Report
All corrected except the title